# Distal Radius Fracture with Ipsilateral Elbow Dislocation: A Rare but Challenging Injury

**DOI:** 10.3390/jpm12071097

**Published:** 2022-07-01

**Authors:** Henrik C. Bäcker, Kathi Thiele, Chia H. Wu, Philipp Moroder, Ulrich Stöckle, Karl F. Braun

**Affiliations:** 1Department of Orthopaedic Surgery and Traumatology, Charité Berlin, University Hospital Berlin, 10117 Berlin, Germany; kathi.thiele@charite.de (K.T.); philipp.moroder@charitee.de (P.M.); ulrich.stockle@charitee.de (U.S.); karl.braun@charitee.de (K.F.B.); 2Department of Orthopaedics & Sports Medicine, Baylor College of Medicine Medical Centre, Houston, 77030 TX, USA; wu.chia.h@gmail.com; 3Department of Trauma Surgery, Technical University Munich, Klinikum Rechts der Isar, 81675 Munich, Germany

**Keywords:** distal radius, elbow dislocation, treatment, epidemiology

## Abstract

Distal radius fractures are common and account for approximately 14% to 18% of all adult extremity injuries. On rare occasions, ipsilateral elbow dislocation can be observed additionally. However, this can be missed without careful examination, especially in patients experiencing altered mental status. The aim of this study was to analyze the mechanism, level of injury, demographics, and associated injuries in distal radius fracture with ipsilateral elbow dislocation. Between 2012 and 2019, we searched our trauma database for distal radius fracture with ipsilateral elbow dislocation. All patients older than 18 years old were included. Data on demographics, mechanism of injury, level of energy, and subsequent treatment were collected. A total of seven patients were identified. The mean age in this cohort was 68.7 ± 13.3 years old, and the left side was involved in 71.4% of the patients. Females were affected in 85.7% (*n* = 6/7) of cases, all of whom suffered from low-energy monotrauma at a mean age of 71.5 ± 12.3 years old. One male patient suffered from high-energy trauma (52 years old). Mainly, posterior elbow dislocations were observed (66.7%; *n* = 4/6). Distal radius fracture patterns, in accordance with the AO classification, included two C2-, two C3-, one C1-, and one B1-type fractures. In the patient suffering from high-energy trauma, the closed distal radius fracture was classified as type C3. Associated injures included open elbow dislocation, ulnar artery rupture, and damage to the flexor digitorum superficialis. Although distal radius fracture with ipsilateral elbow dislocation is thought to be from high-energy trauma, this study shows that most patients were elderly females suffering from low-energy mechanisms. It is important for clinicians to maintain a high level of suspicion for any concomitant injury in this population.

## 1. Introduction

Isolated distal radius fractures are one of the most common injuries and account for between 14% and 18% of all adult extremity injuries [1,2]. Risk factors include osteoporosis, White race, and female sex. Osteoporosis has been diagnosed in 64% of patients following screening [3]. Furthermore, most patients suffer from a fall in the winter months related to slippery walking conditions [4]. For treatment, a large variety of techniques have been described.

When looking for elbow instability, this is the second most common dislocated major joint after the shoulder [5,6]. It commonly occurs in young male patients with an odds ratio of 1.7–1.8:1 [7]. Most patients are under the age of 30 years [8,9], with peak incidence occurring between 5 and 25 years of age [5]. However, this pattern is different in the elderly population where more women are affected [10]. The dislocation is classified according to the direction of the ulna dislocation, of which posterior dislocation is the most common, seen in up to 79% of cases [11,12].

In patients who fall on outstretched arm, the loads are typically directed through either the distal radius alone or the elbow. Therefore, combined injuries are rare, and few case reports have been described [13,14]. For distal radius fractures, the wrist is typically in dorsiflexion, where the position of the wrist at the moment of impact determines the fracture pattern and concomitant injuries, if any. Pronation, supination, and abduction determine the direction of the force transmission [15]. Similarly, elbow dislocations typically occur from falling onto the extended arm [11] [16]. For posterior elbow dislocation, which is the most common dislocation type (>80% of cases), the injury mechanism is typically described as a combination of axial compression and valgus stress with the forearm supinated and elbow flexed [17,18,19]. For anterior elbow dislocation, the mechanism of injury is an anterior directed force on the proximal ulna with the elbow flexed in most cases [20].

The aim of this study was to investigate the epidemiology, demographics, diagnostic modalities, and subsequent treatment options for distal radius fractures with ipsilateral elbow dislocation at a major level 1 trauma center.

## 2. Materials and Methods

For this retrospective trial, we searched our trauma database between 2012 and 2019 for patients suffering from distal radius fracture with ipsilateral elbow dislocation. Patients older than 18 years of age presenting with a distal radius fracture and ipsilateral elbow dislocation in our emergency department were included. All patients suffering from any elbow fracture were excluded. Clinical records and radiographies were analyzed by a fellowship-trained orthopedic trauma surgeon. Data on age, gender, mechanism of injury, diagnostic modalities, neurologic deficits, fracture pattern of the distal radius, and any subsequent treatments rendered were recorded. Patients were classified according to the mechanism of injury including low (e.g., ground level fall) and high (e.g., motorcycle crash or fall from height). Furthermore, conventional radiographs and computed tomography images were analyzed for fracture patterns. Therefore, the AO/OTA classification was used to classify distal radius fractures. Elbow dislocations were classified according to the direction of dislocation. Information on any other concomitant injuries and comorbidities was noted.

For statistical analysis, Microsoft Excel (Microsoft Corporation, Redmond, WA, USA) and SPSS version 22 (IBM, Armonk, NY, USA) were used. For normally distributed values, the mean and standard error of the mean were calculated. 

## 3. Results

In total, nine patients were found when searching our database between 2012 and 2020. There were two patients who suffered from elbow fractures and, as such, were excluded, leaving seven patients for final inclusion. The mean age was 68.7 ± 13.3 years old, ranging from 52 to 89 years old. Females consisted of 85.7% of cases (*n* = 6/7), and the left side was affected in 71.4% (*n* = 5/7) of cases (Table 1).

Most patients in this study presented with low-energy trauma in 85.7% of cases (*n* = 6/7). In one patient, a high-energy mechanism was described, suffering from a fall from approximately 4 *m* in height (13 feet). Patients presenting with a low-energy mechanism suffered from monotrauma, whereas the one patient who sustained a high-energy mechanism sustained multiple injuries.

### 3.1. Low-Energy Injuries

The mean age of the patients with low-energy injuries was 71.5 ± 12.3 years old. Concerning distal radius fracture patterns, C types were observed in 83.3% of cases (*n* = 5/6), including two type C2, two type C3, one type C1, and one type B1. Furthermore, one patient suffered from an open distal radius fracture. Two patients sustained radiocarpal dislocation. No other concomitant injuries, such as neurovascular injuries, were observed. In all patients, elbow reduction was performed in the emergency room. Two patients underwent external fixation as initial treatment. In one case, external fixation was indicated for both elbow and wrist, and in another patient, it was only indicated for the wrist. For the remaining patients, the elbow was stable once reduced. Surgery was indicated in five patients consisting of a volar plate osteosynthesis, and in one patient addition suture anchors were required in order to stabilize the elbow. Conservative treatment was indicated in the nondisplaced B3-type distal radius fracture. This was treated with casting. In patients with stable elbow, the elbow was immobilized for one week, followed by a limited active range of motion in a hinged elbow brace allowing 20° to 90° of flexion for three weeks. Afterwards, patients were allowed a range of motion as tolerated without lifting of heavy weights for another two weeks. After operative treatment of the distal radius, active mobilization was allowed from day one while maintaining non-weight bearing. In all patients, a good to excellent range of motion (0°/0°/150°) of the elbow or wrist was achieved after intensive physiotherapy. 

### 3.2. High-Energy Injuries

Only one male patient suffered from isolated distal radius fracture and elbow dislocation. At the time of accident, he was younger than the average age in the low-energy group. The left side was affected, and an open elbow dislocation was noted with an ulnar artery disruption. Other injuries included a pneumothorax on the left side, several rib fractures, spinopelvic dissociation type II, acetabulum fracture, and five lumbar transverse process fractures. Initially, damage control surgery was performed. This included temporary external fixation of the elbow and wrist, repair of the ulnar artery, local debridement, and application artificial skin substitute on the open wound. Once clinically stable, a combination of a volar plate and mini frag plate was used to fix the distal radius and distal ulnar fracture, respectively. For the elbow, lateral ulnar collateral ligament (LUCL) reconstruction was performed using two suture anchors. Postoperatively, the patient showed a limited range of motion in the elbow and the wrist. All injuries are illustrated in Table 2. In addition, one example is shown in Figure 1.

## 4. Discussion

This study showed that combined distal radius fracture and ipsilateral elbow dislocation is most commonly sustained by elderly patients over the mean age of 70 years with a low-energy mechanism. In those with altered mental status, clinicians need to maintain a high degree of clinical suspicion to avoid missing concomitant injuries. Careful history and physical examination is imperative. 

The existing literature tends to focus on isolated distal radius fractures or diaphyseal radial versus ulnar fractures with elbow dislocation [21,22]. There is relatively little published on distal radius fractures with ipsilateral elbow dislocation as it is uncommon. To our knowledge, our paper represents the largest cohort that has been published to date. The most commonly described injury mechanism includes axial loading on an outstretched hand with a supinated forearm and slightly flexed elbow [2]. The distal radius breaks initially due to the direct contact with the ground. Further, forces may transmit through the elbow, causing posteromedial dislocation [14]. Posteromedial dislocation is associated with axial loading of the elbow in a varus position and the forearm in pronation. Specific to our cohort, patients predominantly suffered from a directly posterior elbow dislocation. 

While elbow dislocations are more frequently described in young male patients [7,9], most patients in our cohort were elderly females suffering from low-energy trauma. This may be due to the fact that we excluded any elbow fracture dislocation in our cohort. In a few cases, it is possible that a dislocation spontaneously reduced prior to presentation due to the fact of muscle contraction. It is also possible that the elbow unknowingly reduced when attempting closed reduction of the distal radius fracture. A thorough history and physical examination is crucial in patients where a concomitant injury is suspected.

While the use of plain film and computed tomography to assess fracture morphology is widely accepted, there is no consensus regarding the use of dynamic fluoroscopy as well as MRI to assess instability of the elbow joint. In particular, dynamic fluoroscopy allows detection of subtle instability throughout range of motion. MRI may allow assessment of the quality of soft tissue but does not allow assessment of how the tissue functions under stress. Surgical intervention is not routinely recommended for simple elbow dislocation. It is performed only if there is instability post-reduction, most commonly seen in extension.

Initially management of this combination of injuries typically involves closed reduction of the elbow and distal radius. Intravenous sedation is recommended to avoid patient duress and muscle relaxation while performing the reduction. Elbow stability should ideally be assessed under fluoroscopy in flexion and extension, valgus and varus, as well as pronation and supination [23]. In some cases, recurrent dislocation or subtle instability may be present in extension [16]. This is followed by closed reduction of the wrist, being mindful not to further traumatize the elbow. The elbow and wrist are then stabilized in a posterior long arm splint in 90 degrees of flexion. 

In posterolateral elbow dislocations, the forearm should be pronated, whereas in posteromedial dislocations, it should be supinated when splinting. Subtle instability only present in extension can often be treated in a hinge elbow brace. This allows immobilization of the elbow in 90 degrees of flexion that can be advanced gradually to full extension. In the absence of instability post-reduction, early mobilization should be initiated to maintain functionality. This includes active assisted exercises starting in week 2 after injury. In patients where there is associated elbow fracture or gross instability, surgery is recommended to restore stability [24,25]. 

In our cohort, only two patients required surgical intervention: one suffered from an acute high-energy injury and one from a low-energy injury. For surgical stabilization, repair of the lateral ulnar collateral ligament (LUCL) was performed after several debridements of the elbow. In the second patient, repair of the capsule, medial, and lateral collateral ligaments were necessary to achieve stability. However, in the remaining five patients, conservative treatments were performed without any chronic instability. 

## 5. Conclusions

Distal radius fracture with ipsilateral elbow dislocation without fracture is a rare type of injury. In our cohort, this most commonly affected elderly patients with a low-energy mechanism of injury. The most common type of dislocation was direct posterior, and most could be treated with closed reduction without surgical stabilization. Any subtle instability should be accessed via dynamic fluoroscopy. Surgical stabilization is recommended for gross instability. Physicians must be wary of any elbow pain in patients who present with distal radius fractures. A careful history, physical exam, and relevant imaging is imperative. 

## Figures and Tables

**Figure 1 jpm-12-01097-f001:**
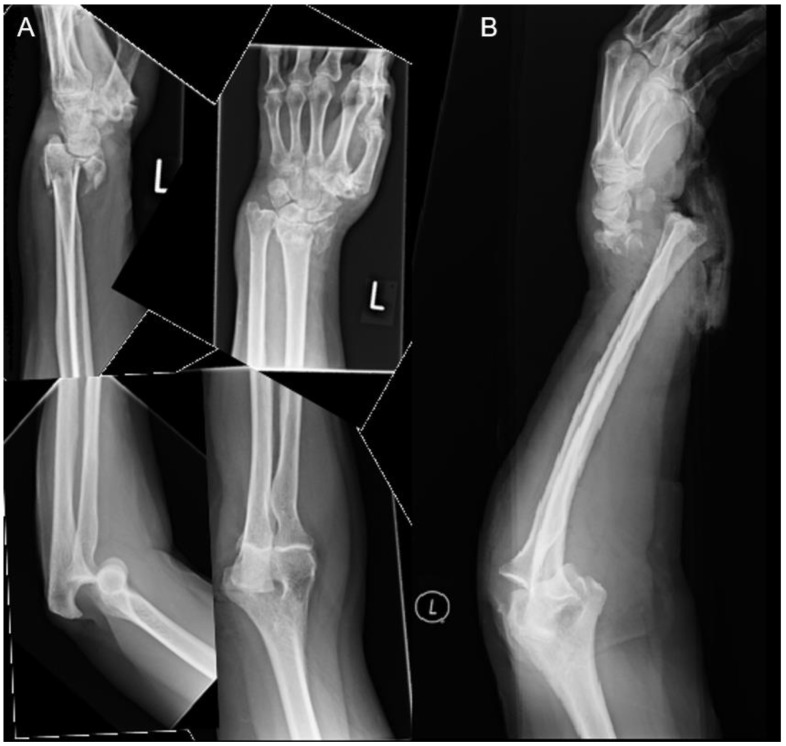
Female patients after low-energy falls with (**A**) a distal radius type-C3 fracture with posterior elbow dislocation and (**B**) an open distal radius type-C3 fracture with a concomitant posterolateral elbow dislocation.

**Table 1 jpm-12-01097-t001:** Demographics of the patients suffering from distal radius fractures and ipsilateral elbow dislocation.

	Numbers (%)
Number of patients (*n*)	7 (100)
Gender (female)	6 (85.7)
Age (years)	65.3 ± 15.4
Level of energy (low energy)	6 (85.7)
Side (left)	5 (62.5)

**Table 2 jpm-12-01097-t002:** Differentiation between low- and high-energy-related accidents, diagnoses, and subsequent treatments.

	Number (%)	Gender (Female)	Age (Years)	Distal Radius Fracture	Colles Fracture (%)	Elbow Dislocation	External Fixation	Volar Plate ORIF (%)	Elbow Stabilization (%)	Concomitant Injuries
Low-energy injuries	6 (85.7)	6 (100)	71.5 ± 12.3	1xB1; 1xC1; 2xC2; 2xC3	4 (57.1)	4 post.; 1 post-lat; 1 divergent	1 wrist; 1 elbow	5 (83.3)	1 (16.7)	1 – open fx
High-energy injuries	1 (14.3)	0 (0)	52	1xC3	1 (100)	unclear	1 elbow and wrist	1 (100)	1 (100)	1 – open fx and ulnar artery lesion
Total	7 (100)	6 (85.7)	65.3 ± 15.4					6	2 (28.6)	

## Data Availability

All data are presented in the manuscript.

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
