# Peer review of "Distal Radius Fracture with Ipsilateral Elbow Dislocation: A Rare but Challenging Injury"

_jpm, 2022, doi:10.3390/jpm12071097_

Round 1

Reviewer 1 Report

This is a good study that focuses on relatively rare cases and provides valuable information for clinicians, suggesting language changes for publication

Author Response

Thank you so much. We edited the manuscript entirely and asked our native speaking colleague to proof read it.

Reviewer 2 Report

Authors should feel encouraged to ask for support some fluent English-speaking person, who would help them to fix several linguistic pitfalls. An appropriate commas put into the text would help it's understanding. 

[51-62] "...In patients who fall on the outstretched arm, the forces are typically directed through either the distal radius alone or the elbow..." I would suspect that a hand fingers could be outstretched, but arm adducted or abducted, flexed or extended, rotated interiorly or exteriorly. In this sentence the word "loads" ionstead of "forces" seem to fit the problem better.

[58] once again "...outstretched arm..."

[58-64] the described mechanism of various types of elbow dislocation are not easily understood.  How could one imagine valgus stress of the flexed elbow?  External rotation of the shoulder?

[77-78] "...The level of energy was divided into..." could be explained as, i.e.,  "... patients were classified according to the estimated energy of the injury..."

[90] Authors could consider the use of term "in consequence" instead of "after" in sentence "...elbow dislocation fractures after..."

[130] ribs instead of "...costae fractures.." seems to soud better

[133] "...sythetic skin replacement..." - do Authors mean covering with the artificial skin substitute?

[135-6} "...fixation of the ulnar through the ulnar approach..." what was fixed  - elbow or ulna? Do Authors mean stabilization of the fractured distal ulna or dislocated elbow? How it was fixed / stabilized? Were any implants used? 

[150] the word "...overlie..." could be replaced by <dominate>.

[159] "...elbow in a hyperextended position..." - do Authors suggest that a range of elbow extension was higher than physiologic one?  Or do they wanted to say "...elbow in fully / completely extended position..."?

[162] The arm could not be positioned in valgus. Valgus and varus positions are possible between bones forming the joint or between fragments of the broken bone. Again, could arm be outstretched?

[170]"..missed..." - the word ,incomplete> may sound better

[171] "...sub-dislocation...". I think Authors mean <sub-luxation> or <incomplete/partial dislocation>

[183] "...re-dislocation..." means <reccurent / secondary dislocation>?

[196] what is "..stable situation..."? How was it achieved?

 [208-9] in adjacent sentences two contrary (rare and typically) terms are used causing some confusion. Nevertheless, I fully agree with Authors that distal radial fractures corresponding with elbow dislocations are rare, andwhen happen, in most cases occur in consequence of low energy injuries. 

would suggest the Authors to 

Author Response

(The authors gave the same response as above.)
